# Specific, sensitive and quantitative protein detection by in-gel fluorescence

Adrian C. D. Fuchs ●[1] ✉

Recombinant proteins in complex solutions are typically detected with tag-specific antibodies in Western blots. Here we describe an antibody-free alternative in which tagged proteins are detected directly in polyacrylamide gels. For this, the highly specific protein ligase Connectase is used to selectively fuse fluorophores to target proteins carrying a recognition sequence, the CnTag. Compared to Western blots, this procedure is faster, more sensitive, offers a better signal-to-noise ratio, requires no optimization for different samples, allows more reproducible and accurate quantifications, and uses freely available reagents. With these advantages, this method represents a promising alternative to the state of the art and may facilitate studies on recombinant proteins.

Western blots present one of the most widely used methods in molecular research. Here, proteins within a complex mixture are separated on a polyacrylamide gel and transferred to a membrane. After additional binding sites on this membrane have been blocked, target-specific antibodies are used to detect the protein of interest (POI), and these, in turn, are visualized by reporter-conjugated antibodies. Overall, the procedure involves multiple steps, which are often adapted depending on the target protein, the chosen antibody combination, or the sample composition[1,2]. This impedes the comparability of different blots, in particular across different labs[3–5]. Even with identical samples and standard operating procedures, results from different users may vary significantly[6]. In addition, quantitative analyses are complicated[4] and often result in a hyperbolic rather than in the desired linear signal-to-substrate relationship[1,4,7].

While commercial POI-specific antibodies are available for many proteins from model organisms and for well-studied targets, no suitable antibodies are on the market for the majority of proteins. In such cases, a small peptide tag can be introduced, which is detectable with a tag-specific antibody. As these antibodies are often better characterized and more reliable compared to POI-specific antibodies[8], many researchers prefer this method to study exogenous recombinant proteins, and an increasing number of scientists use gene-editing tools to insert tags in endogenous proteins[9–11]. Popular tags include Streptavidin-, FLAG-, V5-, cMyc-, His$_6$-, HA-, or E-tags, and vary between 6–14 amino acids in length, plus additional linker sequences. Similar small protein tags can also be detected by other means. For example, a

C-terminal 11 amino acid fragment, the HiBiT tag, can be used to reconstitute a split luciferase in order to visualize the tagged protein on a blotting membrane[12]. Alternatively, His$_6$-[13] or His$_{12}$-tags[14] can be detected directly in polyacrylamide gels by fluorophore-conjugated chelator probes. Remarkably, however, no protein ligase-based system has yet been established, primarily because the few known ligase enzymes have a low substrate specificity and suffer from side reactions[15].

Recently, we identified a protein ligase with very different characteristics[16]. This enzyme, Connectase (Cnt), is found in methanogenic archaea. Here, it recognizes a sequence in its interaction partner Methyltransferase A (MtrA), which links the catalytic and transmembrane domain. This sequence consists of a highly conserved KDPGA motif and 10 residues C-terminal of that motif, which vary across different species. We found that Connectase acts on this recognition sequence, even when it is fused to other proteins or molecules. Upon binding, Connectase cleaves the amide bond between aspartate and proline in the KDPGA sequence and forms a new amide bond between its own N-terminal amino group and said aspartate. In other words, the N-terminal substrate fragment ending in KD is fused to the enzyme (in the following N-Cnt), while the C-terminal fragment starting with PGA plus another 10 amino acids (in the following CnTag) is cleaved off. This reaction is reversible, meaning that the substrate is constantly cleaved and re-ligated. Consequently, an alternative CnTag-substrate can replace the original CnTag-fragment in the reverse reaction, resulting in the formation of a new fusion product (Fig. 1).

---

[1]Department of Protein Evolution, Max Planck Institute for Biology, 72076 Tübingen, Germany. ✉e-mail: adrian.fuchs@tuebingen.mpg.de

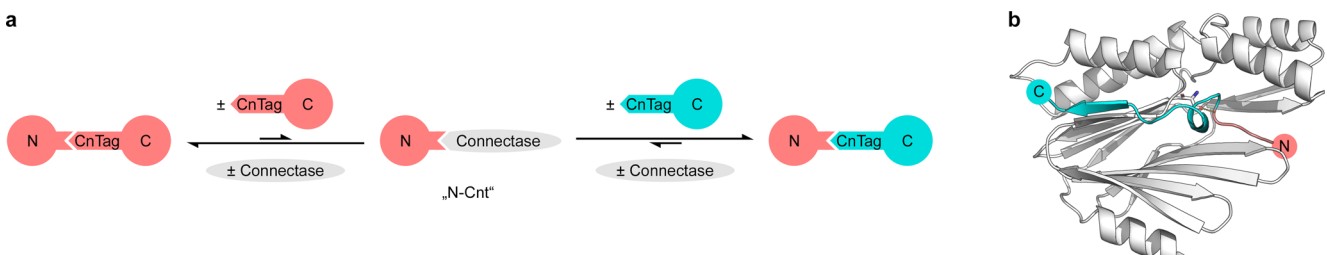

**Fig. 1 | Schematic representation of a Connectase-mediated ligation of two substrates, N and C (a), and AlphaFold model (b) of *M. mazei* Connectase (white) in complex with its recognition sequence (red, cyan).** Connectase forms an amide bond between its active site residue (Thr-1) and the N-terminal part of a substrate (N-Cnt), while the C-terminal part of the substrate is cleaved. The reaction is reversible so that the amide bond in the original substrate (red) can be reconstituted. Alternatively, a new fusion product with an alternative C-terminal fragment (blue) can be formed.

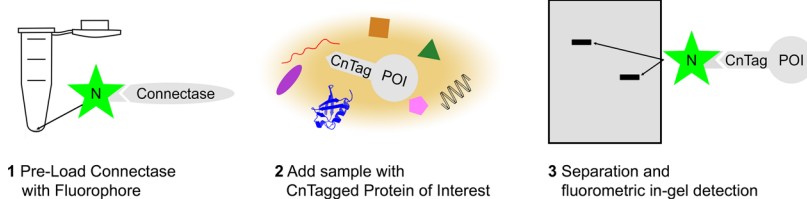

**1** Pre-Load Connectase with Fluorophore

**2** Add sample with CnTagged Protein of Interest

**3** Separation and fluorometric in-gel detection

**Fig. 2 | Schematic representation of the in-gel fluorescence detection method.** Connectase is pre-loaded with a fluorescent peptide substrate (1) and mixed with a sample containing a CnTagged Protein of Interest (2). After a short incubation step, the fluorescent protein of interest can be visualized on a polyacrylamide gel (3).

Compared to known protein ligases, Connectase requires a longer recognition sequence and consequently shows very high specificity and affinity for its substrates. Furthermore, its unique reaction mechanism prevents the usual proteolytic side reactions entirely.

## Results and discussion
### Sensitive and reliable target protein detection

For our experiments, we used Connectase from *Methanosarcina mazei* and performed all reactions at room temperature, neutral pH (7.0–7.5), moderate salt concentrations (150 mM NaCl, 50 mM KCl), and in presence of *E. coli* cell extract to simulate sample impurities. We designed a peptide substrate, with the Cy5.5 fluorophore at the N-terminus, followed by 20 residues based on the *Methanosarcina mazei* MtrA sequence (Cy5.5-RELAS**KDPGA**FDADPLVVEI). Furthermore, we generated target proteins with N-terminal CnTags (PGAFDADPLVVEI) and short linker sequences (5 amino acids, e.g., AAAGA (see Supplementary Information)). These could be obtained by routine cloning and purification procedures because the start-methionine preceding the CnTag (PGA...) is removed during protein expression by endogenous methionine aminopeptidase[16,17].

Using these components, we established a standard protocol to visualize proteins with N-terminal CnTags (Fig. 2; C-terminally tagged proteins require a different procedure and are not studied here). It involves:

(1) the formation of a fluorophore-Connectase conjugate (N-Cnt; see Fig. 1a, middle) by incubating equimolar concentrations (5 μM) of Connectase and the fluorescent peptide substrate for 1 min. The reaction results in an equilibrium, where approximately one out of four Connectase enzymes are conjugated to the fluorophore (i.e., 1.25 μM N-Cnt; Supplementary Fig. 1).

(2) mixing 6.67 nM of this reagent (i.e., 1.67 nM N-Cnt) with the solution containing the protein of interest (POI). This concentration worked for any POI, any concentration and any buffer in this study. The labeling reaction should be incubated for ≥5 min at room temperature in order to obtain a sensitive signal (qualitative analyses) and for 30 min to perform quantitative analyses (Supplementary Fig. 2).

(3) separating the samples on a polyacrylamide gel and analysis on a fluorescence imager or scanner. The signal is stable for several days. For longer storage, the gels should be fixated with 50% methanol/10% acetate. Alternatively, gels can be stained with unspecific dyes (Coomassie brilliant blue) or used for Western blots.

We tested this approach for visualizing small amounts of various CnTagged proteins (125 fmol each, i.e., 1–8 ng depending on the molecular weight of the protein) in large quantities (~20 μg) of cell extract (Fig. 3). In the resulting gel, all target proteins could be detected with a similar signal, while no unspecific bands were seen. Multimeric assemblies (GroES, GST, PsmA, OmpLA[18]) and membrane proteins (OmpLA) could be detected without difficulty. The employed fluorophore-Connectase conjugate N-Cnt could only be detected as a band in a control reaction without CnTagged POI (first lane in Fig. 3). This band disappeared almost entirely when a CnTagged POI was present, indicating an effective transfer of the fluorophore. In sum, these results show that the in-gel fluorescence assay is a general method to detect CnTagged proteins in a complex mixture with good sensitivity and a high signal-to-noise ratio.

We also tested the procedure in buffers used to keep insoluble proteins in the unfolded state (i.e., 4 M urea) and buffers used for denaturing cell lysis (i.e., RIPA buffer, including 1% NP-40, 0.5% deoxycholate, 0.1% SDS). Labeling rates were only slightly decreased at 4 M urea (Supplementary Fig. 3), so that inclusion body proteins in urea may be detected with the same procedure as soluble proteins in physiological buffer. Even more surprisingly, we found that denaturing RIPA buffer increases labeling rates (typically two-fold) compared to detergent-free buffers. Consequently, cell lysates can be used directly for in-gel fluorescence detection.

In the next step, we analyzed the signal-to-substrate relationship in such reactions. For this, we used a serial dilution of CnTagged proteins in constant amounts (~20 μg) of cell extract. A densitometric analysis of the resulting gels (Fig. 4) shows similar sigmoidal signal increases with increasing target protein quantities (on a log scale). The maximum signal is detected at POI quantities >25 fmol (750 pg in case of a 30 kDa protein) and the half-maximum signal is observed at

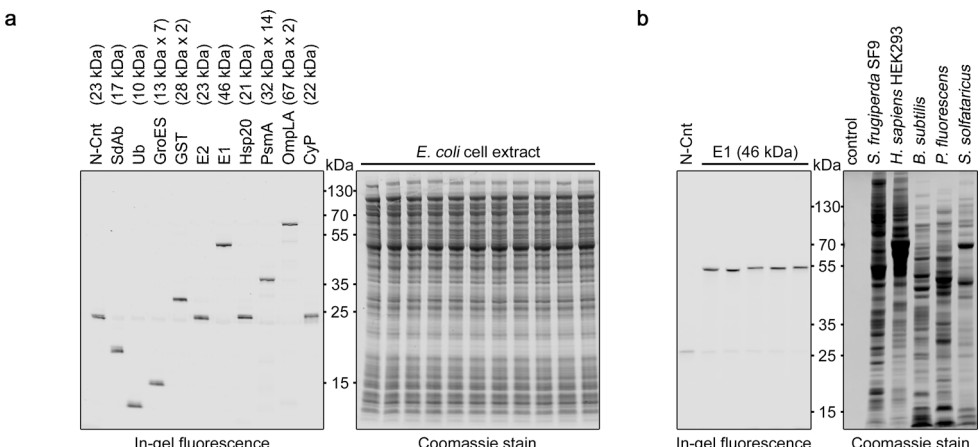

**Fig. 3 | Detection of CnTagged proteins in cell extract via in-gel fluorescence and Coomassie stain of the same SDS-gels. a** Small quantities (125 fmol) of CnTagged proteins were mixed with large quantities (~20 μg) of *E. coli* cell extract and visualized via in-gel fluorescence (left) or with Coomassie (right). The shown proteins are Single domain Antibody (SdAb), Ubiquitin (Ub), GroES, Glutathione-S-Transferase (GST), Ubiquitin-conjugating enzyme (E2), Ubiquitin-activating enzyme (E1), Heat-shock protein 20 kDa (Hsp20), Proteasome subunit Alpha (PsmA), Outer membrane phospholipase A1 (OmpLA) and Cyclophilin A (CyP). The molecular weights and native assembly states of the proteins are indicated. The

employed fluorophore-Connectase conjugate (N-Cnt, see control in lane 1) is only visible as a very faint band in presence of CnTagged POI, indicating an effective transfer of the fluorophore. In high-percentage gels, residual fluorescent peptide substrate (~3 kDa) can be seen at the bottom of the gel (see Supplementary Information). Source data are provided as a Source Data file. **b** Small quantities (125 fmol) of CnTagged E1 protein were mixed with large quantities of *Spodoptera frugiperda* SF9, human HEK293, *Bacillus subtilis*, *Pseudomonas fluorescens* or *Sulfolobus solfataricus* cell extracts and visualized via in-gel fluorescence (left) or with Coomassie (right). Source data are provided as a Source Data file.

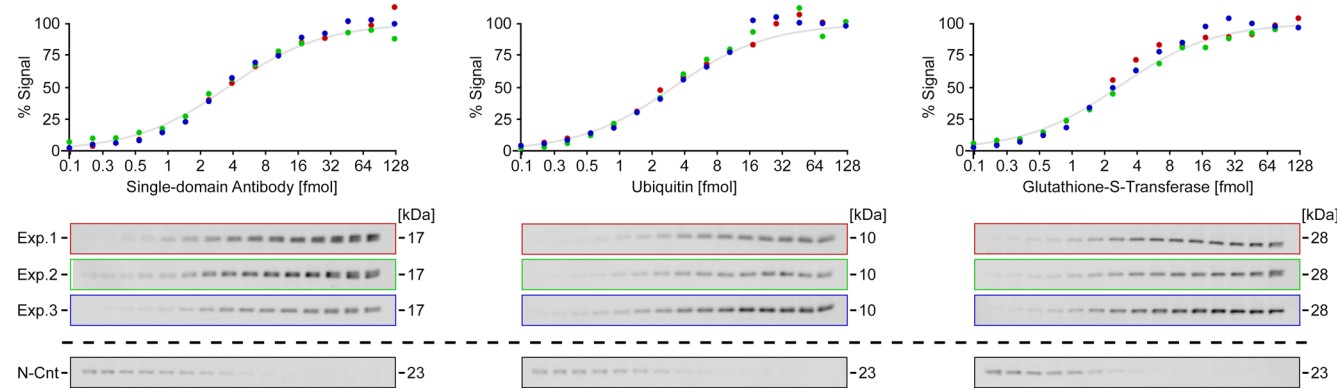

**Fig. 4 | Signal-to-substrate relationship in in-gel fluorescence assays.** A serial dilution (from right to left) of CnTagged proteins was detected via in-gel fluorescence, and the normalized densitometric data was plotted. For each protein, three

independent experiments (red, green, blue; *n* = 3) were conducted. The band corresponding to residual N-Cnt is shown for one representative experiment. Source data are provided as a Source Data file.

~3 fmol (90 pg), suggesting that all pre-formed N-Cnt (~6 fmol) reacts almost quantitatively with the target proteins. In accordance, excess N-Cnt reagent can be detected as an extra band at lower POI levels.

We then compared this approach to Western blots performed under similar conditions (i.e., same cMyc-tagged proteins, IRDye 680-linked secondary antibody, same fluorescence scanner). For this, we transfected adherent HEK293 (*H*uman *E*mbryonic *K*idney) cells with plasmids encoding for cMyc- or CnTagged proteins, lysed them in RIPA buffer, and analyzed different cell extract quantities. In Western blots, the target proteins were detected with varying sensitivity and a detection limit of roughly ~500 ng cell extract protein (Fig. 5a). In in-gel fluorescence assays, the target proteins were detected with uniform sensitivity and a detection limit of ~5 ng cell extract protein. To rule out differences in protein expression, we repeated the assay with a serial dilution of purified cMyc- or CnTagged proteins in constant quantities of *E. coli* cell extract (20 μg per lane; Fig. 5b). Here, the proteins were detected with a sensitivity of ~100 fmol (3 ng of a 30 kDa protein) in the Western blot and ~0.1 fmol (3 pg) in the in-gel fluorescence assay. It

should be noted, however, that Western blot sensitivities are highly variable depending on the detection method, the experimental procedure, and the antibody–POI pairs. For example, published fluorescent blots have often lower (~10 fmol[1,19]) and rarely higher (~1000 fmol[20]) detection limits than those reported here. Chemiluminescent blots can be more sensitive in our experience (some studies disagree[7,20,21]) but are not state of the art for quantifications[1,4,21]. Other in-gel detection methods have published detection limits of 100–200 fmol[13,14] (Supplementary Fig. 4). Overall, it can be concluded that in-gel fluorescence assays will in most cases be one or several orders more sensitive than comparable Western blots or other in-gel detection methods.

In-gel fluorescence also proved a more reliable detection method compared to Western blots with tag-specific antibodies. In our experiments, we obtained relatively strong signals for cMyc-tagged Actin, E1, GFP, CyP, GST, and SdAb proteins, but very weak signals for Hsp20, Ubiquitin, and GroES (Fig. 5a and following sections). A study[22] on the employed monoclonal 9E10 Anti-cMyc antibody (>7000 cites

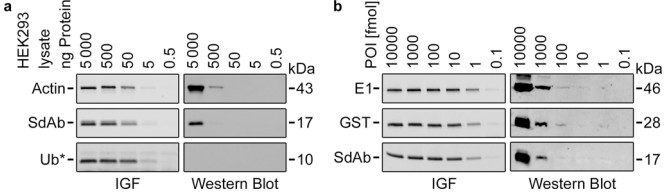

**Fig. 5 | Sensitivity of Western blot and in-gel fluorescence (IGF) experiments.**
**a** Three CnTagged or cMyc-tagged proteins (Actin, SdAb, Ub) were expressed in
HEK293 cells. Lysate samples containing the indicated protein quantities were
analyzed via in-gel fluorescence or Western blot. (* Ub was detected as a monomer
and in conjugates (see Source Data file.) **b** Three CnTagged or cMyc-tagged pro-
teins (E1, GST, SdAb) were expressed in *E. coli* and purified. Samples containing the
indicated molar protein quantities were analyzed via in-gel fluorescence or Western
blot. In-gel fluorescence bands for POI quantities above 10 fmol are equally intense
due to the limited amount of used reagent (~6 fmol N-Cnt; see also Fig. 4). Source
data are provided as a Source Data file.

according to www.citeab.com) confirmed this context dependency.
The authors permuted the first two amino acids C-terminal of the
cMyc-tag in peptide microarrays and found that 122 out of 800 var-
iants were detected with <10% signal intensity compared to the aver-
age; 20 variants could not be detected. A second study[23] showed that
some cMyc-tagged proteins can be detected in immunofluorescence
and immunoprecipitation assays, but not in Western blots, suggesting
that the structure of the (partially) denatured protein on the mem-
brane affects tag accessibility or binding affinity. A strong variability
was also described for the detection of other small protein tags[24]. In
contrast to this, we observed basically the same labeling behavior for
all CnTagged proteins (16 in this study; see also Figs. 4 and 5).

The method presented so far is sensitive and reliable, but, unlike
the Western blots in Fig. 5, it gives no quantitative information. This is
because Western blots typically use an excess of antibodies to detect
relatively smaller POI quantities, while in-gel fluorescence assays use a
limited amount of fluorophore to detect relatively larger POI quan-
tities. Fluorophore limitation brings many advantages: The same pro-
tocol can be applied for every sample, irrespective of how much target
protein is contained. In every case, a noise-free gel image with a
defined band and no danger of overexposure or auto-quenching will
be obtained. Crucially, however, fluorophore limitation allows a more
accurate way of quantifying proteins, which will be presented in the
next section.

## Relative and absolute protein quantifications

When two CnTagged proteins are used in the in-gel fluorescence assay,
they can be expected to compete for the limited N-Cnt reagent. With
that in mind, we asked how the observed signal ratio between two
competing proteins relates to their relative abundance. If a given
amount of a target protein (POI) returns a band that is just as bright as
that of a second reference protein ($Signal_{POI}/Signal_{Ref} = 1$), would it
then return twice as much signal if it was twice as concentrated
($Signal_{POI}/Signal_{Ref} = 2$)? This idea can be summarized as

$$Substrate\ ratio \sim Signal\ ratio$$

or

$$\frac{[POI]}{[Ref]} \sim \frac{Signal_{POI}}{Signal_{Ref}}$$

or

$$[POI] \sim Signal\ ratio \times [Ref] \qquad (1)$$

[Eq. (1): Relative quantification]

To investigate this, we designed a competition experiment
(Fig. 6a), where we mixed constant quantities of one reference protein
(125 fmol) with varying quantities of the protein to be analyzed (e.g.,
3.906–3906 fmol). The resulting band intensities were quantified
densitometrically and analyzed in plots (Fig. 6b and Supplementary
Fig. 5). Here, we observed a strictly linear correlation between POI
quantities and signal ratio, suggesting the general validity of the above
equation. However, the curves were shifted in *Y*-direction relative to
each other, that is, either upwards (e.g. Hsp20 (Fig. 6c)) or downwards
(e.g. Ub (Fig. 6b); on a linear scale the curves have different slopes
instead (Supplementary Fig. 6)). This means that Hsp20 returns more
signal than Ub when compared to the same amounts of reference
protein. That should not come as a surprise: After all, both proteins can
be expected to have a different reactivity and to affect the brightness
of the attached fluorophore in different ways (quenching[25]). The sum
of these effects can be summarized by the protein-pair specific con-
stant *k* in the equation

$$[POI] = \frac{Signal\ ratio \times [Ref]}{k} \qquad (2)$$

[Eq. (2): Absolute quantification]

In summary, this means that the competition assay allows relative
protein quantifications when the constant *k* is unknown (Eq. (1)), and
absolute protein quantifications when *k* is known (Eq. (2)). This is
illustrated in Fig. 7. Here, a relative quantification (step 3A) is per-
formed simply by comparing the signal ratios of different samples. In
this case, it can be inferred that sample 2 contains 1.65× more POI
compared to sample 3 (1.35/0.82 = 1.65) and sample 1 contains 2.49×
more POI compared to sample 3 (2.12/0.82 = 2.49).

When a sample with known POI quantities is available (Fig. 7, step
3B), it is possible to determine *k*. For this, it is sufficient to measure the
signal ratio of any POI/reference mixture of known concentration (see
Eq. (2); several measurements are advisable for higher accuracy). In the
simplest case, a 1:1 mixture is used so that Eq. (2) is reduced to

$$Signal\ ratio = k \qquad (3)$$

[Eq. (3): Determination of *k* with [POI] = [Ref]]

This equation also clarifies the biological meaning of *k*: it compares
the signals obtained from the same amounts of POI and reference. The
so-determined *k*-value can then be used in any subsequent experiment
to calculate unknown POI quantities. In Fig. 7, *k* = 1.27, and the known
amounts of reference protein (125 fmol) are used to calculate that
sample 1 contains 245 fmol POI (2.12 × 125 fmol/1.27 = 245 fmol),
that sample 2 contains 162 fmol (1.35 × 125 fmol/1.27 = 162 fmol), and
that sample 3 contains 98 fmol (0.82 × 125 fmol/1.27 = 98 fmol).

## Accuracy, reproducibility, and comparability of results

In the next step, we determined the accuracy of such quantifications.
For this, we calculated protein quantities via Eq. (2) in 12 independent
experiments with a total of 165 signal ratio measurements. We then
compared the results to the (known) POI quantities that were actually
used in the experiments (Fig. 6 (bottom layer) and Supplementary
Fig. 5). We found that the accuracy of the quantification depends on
the signal ratio: Equally intense bands can be compared more accu-
rately, while for bands with very different intensity, small errors in the
quantification of the less intense band lead to high errors in the cal-
culated POI quantities. Based on these results, we suggest to use the
assay at signal ratios between 0.1 and 10, which is sufficient to quantify
samples with a 100× difference in POI levels.

Overall, the method proved very accurate with an average error
of 4.9% in the suggested signal ratio range (Fig. 6 (bottom layer) and
Supplementary Fig. 5). Notably, this error includes both technical
errors, such as pipetting errors, and possible deviations from the

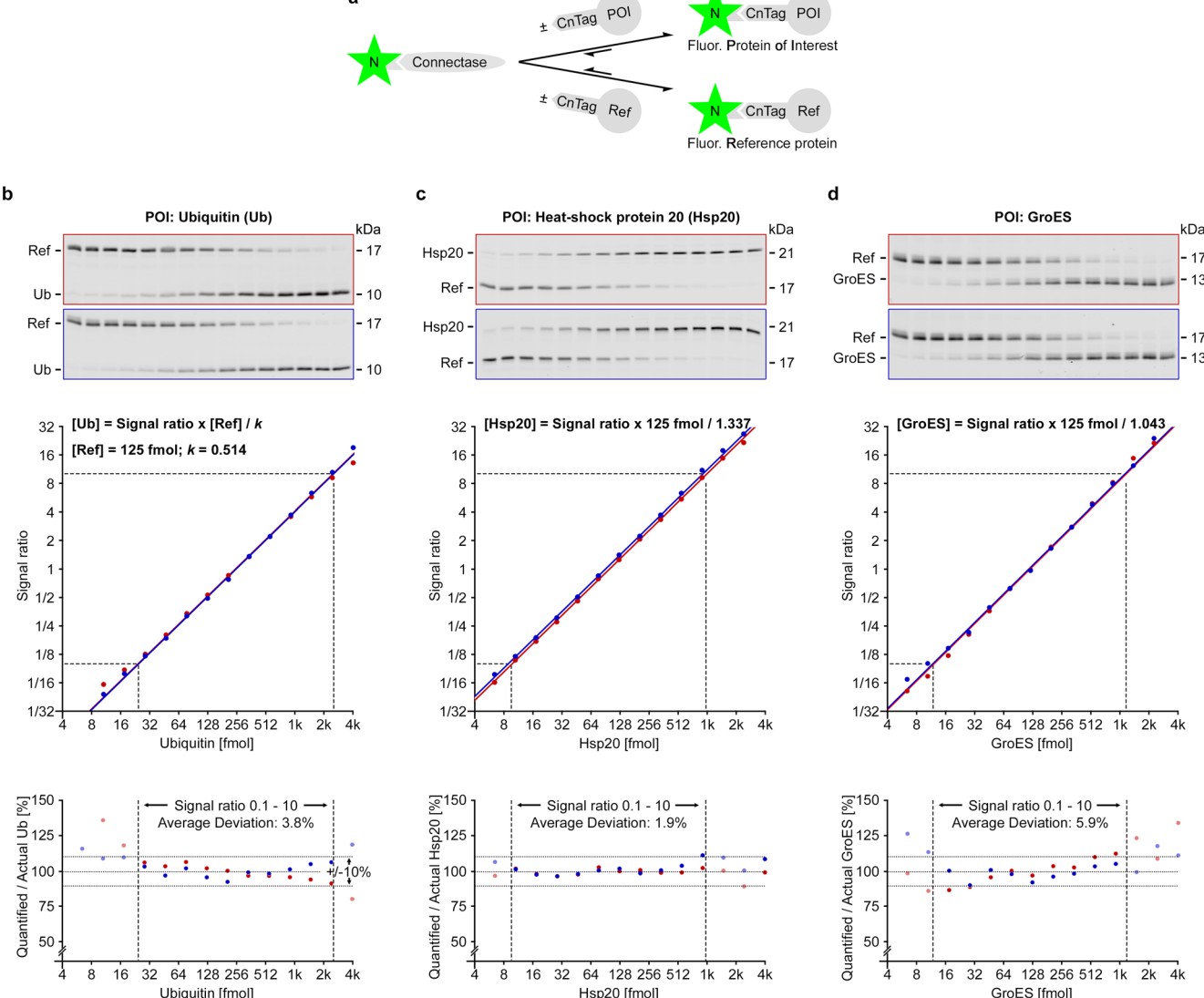

**Fig. 6 | Protein quantification with the in-gel fluorescence competition assay.**
**a** Two CnTagged proteins, the reference (Ref) and the protein of interest (POI), compete for fluorophore-conjugated Connectase. **b–d** Three competition experiments ($n = 2$ replicates colored red and blue) with constant amounts (125 fmol) of a reference protein (SdAb) and increasing quantities of a protein of interest (Ub (**a**), Hsp20 (**b**), GroES (**c**)). The reactions were separated on polyacrylamide gels (top layer), allowing the densitometric quantification of fluorescent protein bands. The resulting data is represented in plots (middle layer; the same plots on a linear scale are shown in Supplementary Fig. 6), revealing a linear relationship between signal ratio and protein of interest quantities in all cases. The individual curves are shifted in $Y$-direction by the factor $k$ (see Eq (2); $k_{(Ub/SdAb)} = 0.518$ (Exp. 1) and 0.509 (Exp. 2; the average of both values is shown in the plot); $k_{(Hsp20/SdAb)} = 1.272$ (Exp. 1) and 1.402 (Exp. 2); $k_{(GroES/SdAb)} = 1.024$ (Exp. 1) and 1.061 (Exp. 2)). The linear relationship allows the quantification of target proteins in unknown samples (see Fig. 7). The error of such quantifications (i.e., the deviation from Eq. (2)) is represented in plots (bottom layer). The most accurate values are obtained at signal ratio 0.1–10 (dotted lines). Source data are provided as a Source Data file.

presumed linear signal ratio–substrate quantity relationship in Eq. (2). Consequently, if such deviations from this empiric finding exist, they must be small enough to warrant sufficiently accurate quantifications.

In order to compare these results with the state of the art[4], we designed quantitative Western blot experiments with the same cMyc-tagged POI, a comparable fluorophore (IRDye680), and the same instrumentation. However, as cMyc-tagged Ubiquitin and GroES gave only weak signals (Supplementary Fig. 7), we eventually conducted the experiments with two other cMyc-tagged proteins, SdAb and E1, instead. We also used higher POI quantities (12.5–12500 fmol, corresponding to 0.58 ng–0.58 μg in the case of E1). The obtained signal-substrate curves (Fig. 8) were linear for 12× (SdAb, Hsp20)–32× (E1) POI increases (visualized by dotted lines). Within this range, the quantification values deviated on average by 15.9% from the linear data fits

(solid lines). These results are comparable to other published Western blot quantifications, with linear ranges between <10× (CPARP, COXIV, ACTB, GAPDH[4]; α-Tubulin, β-Catenin[21]; Hsp90, Tubulin[1]), 16× (p38, E-cadherin, ERK1/2, GAPDH)[1], ~20× (Actin)[4] and ~64× (Phospho-β-Catenin)[21], and similar deviations from data fits (see Butler et al.[7] for more examples; errors at lower POI quantities are less visible on the commonly used linear scale (Supplementary Fig. 8)). We conclude that in most cases, in-gel fluorescence quantifications have a broader linear range and are more reliable, reproducible, and accurate.

The factors that lead to the smaller error values observed in in-gel fluorescence quantifications (Fig. 6) are the use of the POI/reference signal ratio instead of the raw POI signal and the smaller number of parameters that influence this readout. While the POI signal alone (see Fig. 4) would be affected by variations in the added N-Cnt reagent, pipetting errors when loading the gel, fluorescence scanner sensitivity

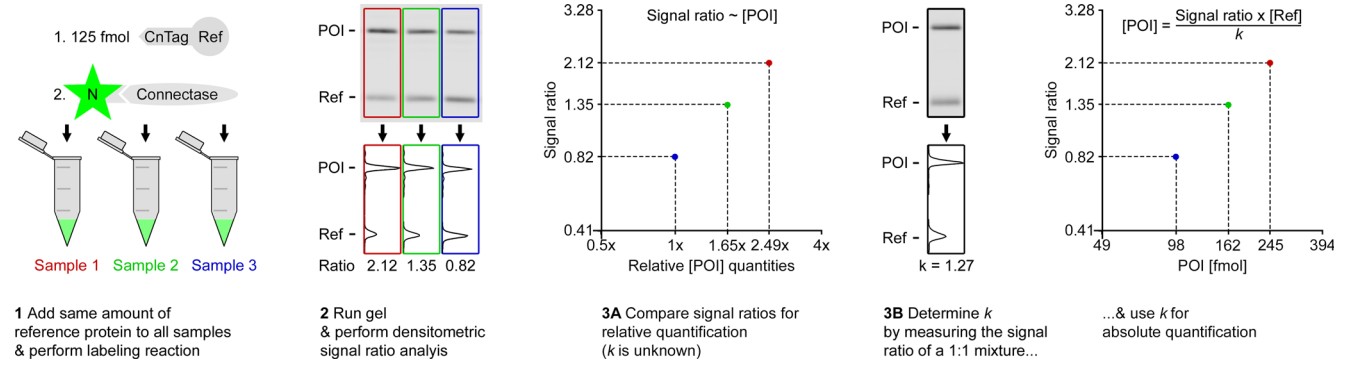

**Fig. 7 | Workflow for relative (3A) and absolute (3B) protein quantification.** Although the results are visualized in plots, it is generally sufficient to use two simple equations for relative (Eq. (1)) and absolute (Eq. (2)) protein quantifications.

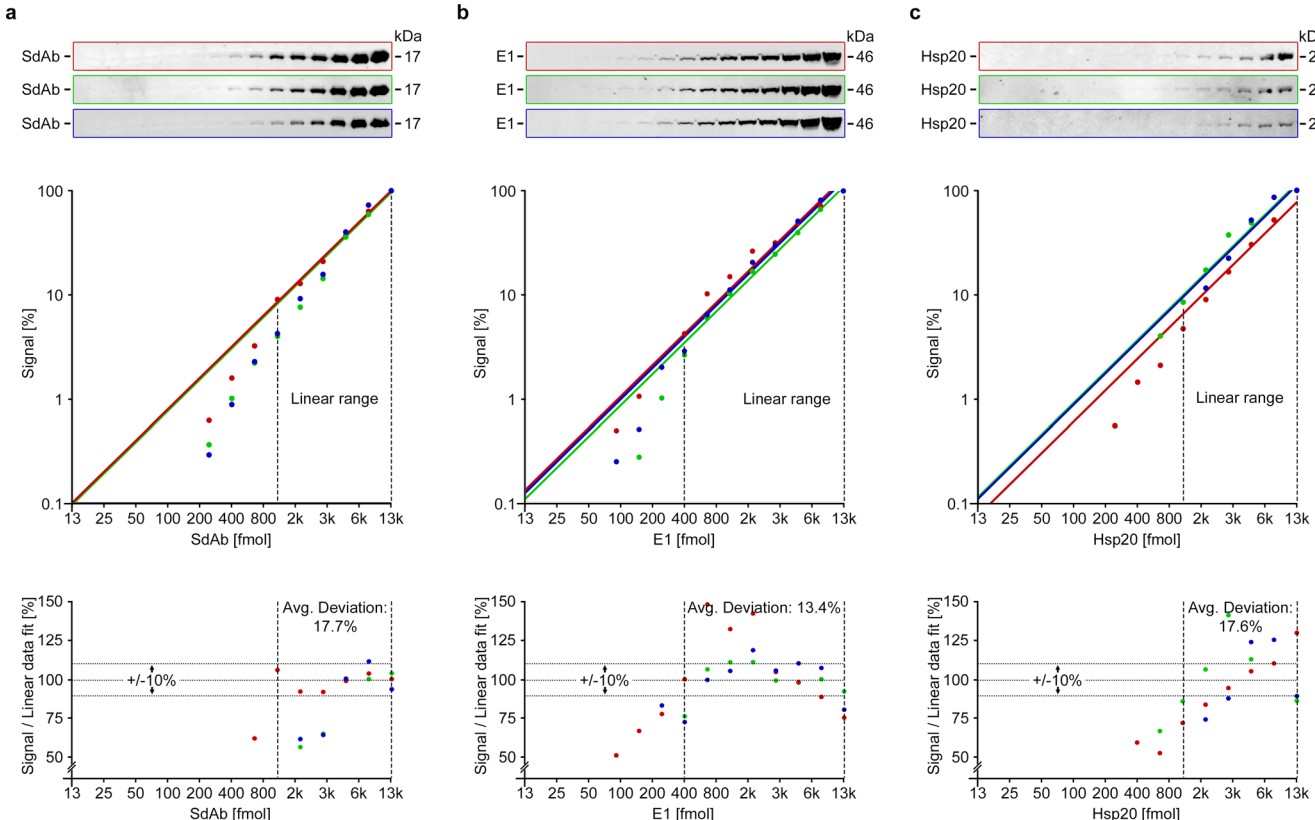

**Fig. 8 | Protein quantification with Western blots. a–c** Shown are three Western blot experiments (n = 3 replicates colored red, green, and blue) with increasing quantities of a cMyc-tagged protein of interest (SdAb (**a**), E1 (**b**) and Hsp20 (**c**); top layer). The resulting POI signals were normalized (100% signal at 12,500 fmol) and plotted against the used POI quantities (middle layer; the same plots on a linear scale are shown in Supplementary Fig. 8). The linear range is indicated by dotted lines and data fits (y = mx) to these linear ranges are indicated by solid lines. The deviation between these fits and the actual data is shown in the bottom layer. Source data are provided as a Source Data file.

settings, or fluorophore bleaching, the POI/reference signal ratio is mostly unaffected by these variables as they influence both the POI and the reference signal in a similar way. Instead, our experiments show that in-gel fluorescence signal ratios are only affected by a few parameters that can be easily controlled: the reaction time (Supplementary Fig. 9), the reaction buffer, and the reaction temperature (Supplementary Fig. 10). By contrast, the POI and reference quantities did not affect the signal ratio, as long as the POI/reference ratio remained constant (Supplementary Fig. 11). Taken together, this means that in-gel fluorescence quantifications show smaller variations than quantitative Western blots, where a lot of parameters affect the POI signals (e.g., transfer, blocking, washing, antibody binding, etc.). With

constant reaction conditions (time, temperature, buffer), the main error source is the densitometric analysis itself (e.g., fuzzy bands, inaccurate background subtraction, etc.). This remaining error is also minimized by the use of the signal ratio: While a 100× difference in POI quantities necessitates the error-prone comparison of very strong and thin bands in Western blots, the analysis of a 10× more intense bands (i.e., a signal ratio between 0.1 and 10) is sufficient in an analogous in-gel fluorescence analysis.

Another consequence of the small number of parameters that affect in-gel fluorescence signal ratios is that the results from independent experiments with (exactly) the same reaction conditions are reproducible (Fig. 6). This suggests that signal ratios from different

gels can be compared if experiments are performed in the same way, which can be an advantage for continuous studies on the same protein of interest. Conversely, raw POI signals from different Western blots can generally not be compared due to unavoidable blot-to-blot variations[1,4] (the blots in Fig. 8 had to be normalized for comparison). For the same reason, it is necessary to load a standard curve with a recombinant protein of known concentration on Western blots for absolute quantifications. As a result, such analyses are rarely seen in the literature. By contrast, a previously determined $k$ value can be used for the same purpose in quantitative in-gel fluorescence analyses, due to the comparability and linearity of the signal ratio read-out (controls are nevertheless mandatory). For example, by mixing the HEK293 cells from Fig. 5a with reference protein, we can conclude in a simple experiment that they contain 603 fmol SdAb per µg cell lysate (Supplementary Fig. 12). We hope that researchers will make ample use of this method in order to obtain more detailed information on biological processes.

## Comparison to alternative methods

The presented in-gel detection method can be used for similar analyses as Western blots with tag-specific antibodies but offers various potential advantages. Data suggests that it is simpler, faster, cheaper, and more sensitive. It is practically free of noise and requires no further optimization—throughout all our experiments, we only varied the analyzed samples but kept exactly the same assay procedure. The utilized gels can be further used for Coomassie-staining (e.g., for normalization) or even Western blotting. Due to the simple procedure, experiments and obtained results become more reproducible, in particular across different labs. The most notable advantage, however, lies in quantifications. Western blots often have a narrow and imperfect linear range[1,4,7]. Conditions that allow quantifications must be identified in previous experiments and are affected by a great number of parameters, such as the blocking agent, band intensities, transfer conditions, or POI quantities/antibody concentrations, to name just a few[1,3,26]. Even seemingly insignificant differences in the procedure may have a surprisingly strong influence on the results[3,4]. Sadly, it is not uncommon in literature to just assume a linear relationship without sufficient controls or validations, resulting in misleading or incorrect interpretations[5,7]. In contrast, the in-gel fluorescence competition assay typically provides a linear signal ratio-to-substrate relationship, enabling accurate relative quantifications with little effort. In addition, it allows absolute quantifications and the comparison of results from different experiments.

It should also be noted that a variety of other in-gel detection methods exist. The company Licor offers protocols for in-gel Westerns, where polyacrylamide gels are soaked in antibody solutions. This procedure is primarily meant for proteins that do not transfer well to membranes, as it requires long incubation and washing steps while offering a lower sensitivity and signal-to-noise ratio compared to regular Western blots. Alternatively, Brüchert et al.[14] have used fluorophore-coupled super-chelators (hexaNTA[Alexa647]) to detect His[12]-tagged proteins on polyacrylamide gels. While the employed tags can also be useful for pulldowns or during protein purification, the method is less sensitive (detection limit 0.2 pmol[14]) compared to the approach presented here (0.1 fmol (Fig. 4 and Supplementary Fig. 4)), and has only been tested for qualitative analyses. Finally, Raducanu et al.[13] pursued a similar approach, except that they used His[6] tags and soaked the gel after the run in staining solution (trisNTA[Alexa647]). This also resulted in a similar detection limit (0.1 pmol) and strongly convex signal-to-substrate curves in quantifications. Taken together, Connectase-mediated in-gel fluorescence offers advantages, not only compared to Western blots with tag-specific antibodies but also compared to existing gel-based methods.

## Future research

Our goal is to establish the in-gel fluorescence assay as a standard method for the detection and quantification of recombinant proteins. As a first step in this process, this paper provides users with a detailed characterization and a general protocol (Supplementary Information). Furthermore, we distribute the required materials for free (see the section "Material availability").

Future research should verify our findings and focus on four goals. First, the method could be improved by using purified fluorophore-conjugated N-Cnt instead of just mixing Connectase and fluorescent peptide (Supplementary Fig. 1). This could result in an increased sensitivity and signal-to-noise ratio as well as reduced labeling times. Second, more efficient and mutually exclusive Connectase/CnTag pairs[27] should be identified. These could be used to detect different proteins in parallel (multiplexing). Third, the method should be adapted to enable the detection of C-terminally tagged proteins and native in-gel fluorescence analysis. And finally, the functionality of the CnTag could be improved, for example by generating CnTag-specific antibodies for pulldowns and affinity purification, or by exploring similar Connectase-based detection methods for microplate readers, cytometry, or fluorescence microscopy.

At the end of this process, we hope to bring a proven and widely applicable protein detection method to the lab and fuel further innovative approaches with this enzyme.

## Methods

### Cloning, expression, and purification

The sequences of all proteins in this study are listed in Supplementary Data 1. The genes were synthesized (Biocat, Heidelberg) or produced via PCR using optimized codon frequencies (E. coli or human). They were cloned into the pET30b(+) vector (restriction sites: NdeI, XhoI) for expression in E. coli or the pcDNA3.1 vector (restriction sites: HindIII, XhoI) for expression in HEK293 cells.

For recombinant expression in E. coli, BL21 gold cells were transfected with the respective plasmids and grown in lysogeny broth medium with 50 µg/l Kanamycin at 22 °C. Protein expression was induced at an optical density of 0.4 at 600 nm with 500 µM isopropyl-β-D-thiogalactoside. Cells expressing soluble proteins were harvested after 16 h, resuspended in buffer (100 mM Tris–HCl, 5 mM MgCl$_2$, 1× cOmplete EDTA-free protease inhibitor cocktail (Roche), 0.02 g/l DNAse, pH 8.0), lysed by the French press, and cleared from cell debris by ultracentrifugation (120,000 × $g$, 45 min, 4 °C). Inclusion body proteins (i.e., PA1732[28]) were cleared from the soluble cell fraction by ultracentrifugation (120,000 × $g$, 45 min, 4 °C), homogenized in washing buffer (1 M urea, 1% Triton, 50 mM Tris–HCl, 250 mM NaCl, 5 mM DTT, pH 8.0), centrifuged a second time (as above), and homogenized in denaturing buffer (8 M urea, 50 mM Tris–HCl, 250 mM NaCl, pH 8.0).

For recombinant expression in human cells, HEK293 cells (DMSZ no.: ACC 305) were cultured in six-well plates with DMEM medium at 37 °C. At 70% confluence, the cells were transfected with 2500 ng of the respective plasmids using 12.5 µl Lipofectamine 2000. The cells were harvested after 24 h.

For protein purification, soluble His$_6$-tagged proteins were applied to HisTrap HP columns (20 mM Tris–HCl pH 8.0, 250 mM NaCl, 20–250 mM imidazole; all columns were obtained from Cytiva) and, in a second step to a Superdex 75 size-exclusion column (20 mM HEPES–NaOH pH 7.5, 100 mM NaCl, 50 mM KCl, 0.5 mM TCEP). His$_6$-tagged inclusion body proteins (i.e., PA1732) were purified with HisTrap HP columns under denaturing conditions (20 mM Tris–HCl pH 8.0, 250 mM NaCl, 20–250 mM imidazole, 8 M urea). All chromatography steps were performed on an Äkta Purifier FPLC (GE Healthcare) using Unicorn v5.1.0 software. Purified proteins were supplemented with 15% glycerol, flash-frozen in liquid nitrogen, and stored at −80 °C.

### In-gel fluorescence assays

Generally, all reactions were conducted as described in the assay protocol (Supplementary Information). Specifically, samples were prepared in buffer A (50 mM sodium acetate, 50 mM MES–NaOH, 50 mM HEPES–NaOH, 50 mM KCl, 150 mM NaCl, pH 7.0) or RIPA buffer (50 mM Tris–HCl, 150 mM NaCl, 1% NP-40, 0.5% deoxycholate, 0.1% SDS, pH 7.5). To avoid protein loss, low protein binding tubes (Eppendorf) or PCR tubes were used and buffer A was supplemented with a carrier protein (0.1 g/l BSA) or *E. coli* BL21 gold cell extract (0.1 g/l protein). Where sample impurities should be simulated, higher levels of cell extract (6 g/l) were used instead (Figs. 3–6, 8, and Supplementary Fig. 2, 5, 7, 9–11). The labeling solution was prepared by mixing equimolar amounts of Cy5.5-RELASKDPGAFDADPLVVEI reagent (synthesized by Genscript; TFA salt; HPLC purity 95.6%; net peptide content 85.9%; molecular weight (2707.16) confirmed via NMR) and Connectase (5 μM each; buffer A without carrier protein). After 1 min at room temperature, this solution was diluted and mixed with the samples to be analyzed (final concentration: 5 nM). The reactions were incubated for 30 min at room temperature and transferred in a fresh tube with 1/4 vol. SDS loading buffer (250 mM Tris, 8% SDS, 0.1% bromophenol blue, 40% glycerol, pH 6.8). The samples were heat incubated (95 °C, 3 min) in some cases (Figs. 3–5, Supplementary Figs. 4 and 12; Note: this step is optional and can be skipped if heat incubation is not required to unfold the target proteins). The samples (5 μl each) were then separated with NuPAGE (Thermo) or mPAGE (Merck) 12% BisTris gels (50 mM MOPS, 50 mM Tris, 1 mM EDTA, 0.1% SDS, pH 7.7; 200 V constant). The gels were imaged immediately after the run with an Odyssey CLx fluorescence scanner (Licor; 700 nm channel, Intensity: Auto, focus offset: 0.5 mm, resolution: 84–337 μm, quality: lowest-medium). 32 bit images containing all fluorescence data (provided in the Source Data file) were generated with the instrument software (Image Studio 5.2.5), exported, and analyzed with ImageJ 1.52a using the Bio-Formats 6.11.0 plugin. For quantifications, the signal of the whole band, including potential smears or shadows was used. For a delayed/later analysis, the gels were stored in a fixation solution (50% methanol, 10% acetate, at 4 °C in the dark; no maximum fixation time). In some cases (Figs. 3 and 6), gels were stained after the fluorescence scan with Coomassie solution (5.8% H$_3$PO$_4$, 10% (NH$_4$)$_2$SO$_4$, 0.12% Coomassie G-250, 20% Ethanol, 5% Methanol), destained (10% acetate), and re-imaged with the fluorescence scanner (same settings). Exceptions from this general procedure and details on each experiment are described in the following.

For the preparation of Fig. 3a, equal quantities of CnTagged proteins (125 fmol SdAb (Single domain Antibody), Ubiquitin (Ub), GroES, Glutathione-S-Transferase (GST), Ubiquitin-conjugating enzyme (E2), Ubiquitin-activating enzyme (E1), Heat-shock protein 20 kDa (Hsp20), Proteasome subunit Alpha (PsmA), Outer membrane phospholipase A1 (OmpLA) or Cyclophilin A (CyP)) in buffer A with cell extract (6 g/l) were visualized as described in the general procedure (see above). For Fig. 3b, various cell extracts were prepared. *Spodoptera frugiperda* SF9 (DMSZ no. ACC 125) and HEK293 cells (DMSZ no. ACC 305) were gifts from Dr. Birte Hernandez Alvarez; *Pseudomonas fluorescens* and *Bacillus subtilis* were cultured in LB at 30 °C up to OD 1.5; *Sulfolobus solfataricus* P2 cells were a gift from Dr. Jörg Martin. All cells were harvested by centrifugation, resuspended in RIPA buffer (supplemented with 0.02 g/l DNAse, cOmplete protease inhibitor, 5 mM MgCl$_2$, and 5 mM TCEP), lysed by sonication, and centrifuged again to obtain the soluble cell extract fraction. The protein concentration was determined with the BCA method. Finally, 25 nM CnTagged E1 were mixed with 4 g/l of each cell extract and visualized as described in the general procedure (see above). The resulting gel showed higher protein quantities for SF9 and HEK293 fractions compared to the *Sulfolobus* fraction, in contrast to the quantities determined with the BCA method.

For Fig. 4, a serial dilution of three individual CnTagged proteins (SdAb, Ub, and GST) was generated, ranging from 50 nM to 50 pM (1.6379× steps; buffer A with cell extract (6 g/l)). These samples were mixed with 1/2 vol. labeling solution and visualized as described in the general procedure (see above). For analysis, the average signal of the 50 nM samples, corresponding to 125 fmol on the gel, was set as 100% signal.

For Fig. 5a, HEK293 cells expressing CnTagged proteins (Actin, SdAb, Ub) were harvested and lysed in RIPA buffer. The protein concentration was determined with Bicinchoninic acid (BCA) and a 10× serial dilution, starting with 2 g/l, was prepared. Likewise, for Fig. 5b, a 10× serial dilution (RIPA buffer with cell extract (6 g/l)) of purified CnTagged proteins (E1, GST, SdAb), starting with 4000 fmol/μl was prepared. All samples were labeled with 1/2 vol. labeling solution and visualized as described in the general procedure (see above).

For Fig. 6, two replicates of three competition experiments were performed. In each case, 1 vol. of a reference protein solution (50 nM SdAb in buffer A with cell extract (6 g/l)) was mixed with 1 vol. of protein of interest solution (2.590–2590 nM (Ub, Hsp20) or 1.581–1581 nM (GroES); 1.6379× steps; same buffer), labeled with 1 vol. of labeling solution, and visualized as described above (final quantities: 125 fmol SdAb; 6.474–6474 fmol Ub or Hsp20; 3.953–3953 fmol GroES (i.e., a shift by one lane (1.6379×) compared to Ub and Hsp20). The signal ratio between reference and protein of interest band intensities was determined with ImageJ and plotted against the protein of interest quantities (middle layer). *k* values were determined for signal ratio 0.1–10 based on Eq. (2) ([POI] = Signal ratio × [Ref]/*k*). The average of these values is shown in the chart and visualized as a linear curve. The percentile deviation between the POI quantities calculated with Eq. (2) and the actually used POI quantities was plotted against the signal ratio (lower layer).

For Supplementary Fig. 1, Connectase and of Cy5.5-RELASKDP-GAFDADPLVVEI reagent were mixed as described above (5 μM each). At the indicated time points (1–1440 min), N-Cnt formation was stopped by transferring an aliquot to a fresh tube with 1/4 vol. SDS loading buffer. The samples were separated via SDS–PAGE. The resulting gel was imaged on a fluorescence scanner, before and after Coomassie staining. The ratio between the resulting bands in the Coomassie-stained gel was plotted against the incubation time.

For Supplementary Fig. 2B, 125 fmol CnTagged E1 protein in RIPA buffer with cell extract (6 g/l) was visualized as described in the general procedure (see above), except that the reaction was stopped at different times (1–1440 min) by transferring an aliquot to a fresh tube with 1/4 vol. SDS loading buffer. For Supplementary Fig. 2C, the reaction was performed with 1–125 fmol CnTagged E1 protein and stopped after 0.2–30 min.

For Supplementary Fig. 3, 1 fmol CnTagged E1 protein was visualized as described in the general procedure (see above), except that the reaction was performed in different buffers and stopped at different times (0.25–30 min) by transferring an aliquot to a fresh tube with 1/4 vol. SDS loading buffer. The chosen buffers were buffer A (labeled Std buffer in the figure), buffer A supplemented with 2–8 M urea, 10% DMSO or 2% SDS, or RIPA buffer.

For Supplementary Fig. 4A, samples containing 45 μM–45 pM CnTagged MBP and 1 g/l *E. coli* cell extract were prepared. Labeling was performed with 1/2 vol. labeling reagent (final concentration 5 nM) and stopped after 30 min by the addition of 1/2 vol. SDS loading buffer. 5 μl of each sample were analyzed on the gel. For Supplementary Fig. 4B, 10 μM–200 pM CnTagged Sumo was used instead of MBP.

For Supplementary Fig. 5, the procedure described in Fig. 6 was repeated, except for the use of different reference (Ub, E2, SdAb, E1) and target proteins (Ub, SdAb, E2, PA1732). The gels show POI quantities ranging in 1.6379× steps from 3.953 to 3953 fmol (POI: Ub/Ref: E2), 2.413–2413 fmol (SdAb/E2), 1.474–1474 (SdAb/Ub) 2.413–2413 fmol

(E2/Ub) and 3.953–3953 fmol (E2/SdAb; E1/PA1732). In the case of Supplementary Fig. 5F, the labeling was performed in the presence of 4 M urea.

For Supplementary Fig. 7, a serial dilution of four individual CnTagged proteins (GST, SdAb, GroES, and Ub) was generated, ranging from 16 μM to 16 pM ($\sqrt{10}$ steps; buffer A with cell extract (6 g/l)). These samples were mixed with 1/2 vol. labeling solution. The labeling reaction was stopped after 30 min by transferring an aliquot to a fresh tube with 1/4 vol. SDS loading buffer. The samples with the same POI quantities (e.g., 16 μM) were then pooled and analyzed via in-gel fluorescence.

For Supplementary Fig. 9, a mixture of two CnTagged proteins (E1/PsmA or GroES/Ub; 125 fmol each in buffer A with cell extract (6 g/l)) was visualized as described in the general procedure (see above), except that the reaction was stopped at different times (1–1440 min) transferring an aliquot to a fresh tube with 1/4 vol. SDS loading buffer. The average $k$ value from two experiments was determined with Eq. (3) and plotted for each time point.

For Supplementary Fig. 10, a mixture of two CnTagged proteins (OmpLA/E1; 125 fmol each in buffer A with cell extract (6 g/l)) was visualized as described in the general procedure (see above), except that the reaction buffer was supplemented with CaCl₂ or EDTA (10 mM), FC-12 (2 mM) or cell extract (6 g/l)/BSA (0.1 g/l). The reaction temperature was set to either 10 °C or 22 °C. The $k$ value for the OmpLA/E1 pair under each condition was determined with Eq. 3.

For Supplementary Fig. 11, a mixture of two CnTagged proteins (E1/PsmA, E2/Ub or SdAb/Ub) in buffer A with cell extract (6 g/l)) was visualized as described in the general procedure (see above), except that different quantities of CnTagged proteins (125–20,000 fmol) were used. The average $k$ value from two experiments was determined with Eq. (3) and plotted against the protein quantities.

For Supplementary Fig. 12, 125 fmol CnTagged E1 was mixed with either 125 fmol CnTagged SdAb or different quantities of SdAb-expressing HEK293 cells (see Fig. 5a) and analyzed as described in the general procedure (see above).

### Western blots

For Western blot analysis (Figs. 5, 8, Supplementary Fig. 7), the samples were prepared as described above (in-gel fluorescence assay, Figs. 5, 6 and Supplementary Fig. 7), except that cMyc-tagged proteins were used and that no labeling reaction was performed. In the case of Fig. 8, cMyc-tagged Hsp20 and E1 proteins were used instead of GroES and Ub (see main text). The samples were heat incubated (95 °C, 3 min), and separated with NuPAGE 12% BisTris gels (Thermo). The proteins were subsequently blotted to Invitrogen Power Blotter Select Nitrocellulose Transfer Stacks (Thermo), using the Invitrogen Power Blotter XL (25 V, 1.3 A, 5 min). The membrane was then blocked (PBST, 5% milk powder, filtered) for 1 h at room temperature and, in the next step, incubated with primary antibody solution (monoclonal mouse anti cMyc 9E10 antibody (Thermo, #13-2500, Lot VK307345; validation: https://www.thermofisher.com/antibody/product/c-Myc-Antibody-clone-9E10-Monoclonal/13-2500), diluted 1:500 (to 1 μg/ml) in PBST) for 16 h at 4 °C. After 5 wash steps (PBST, 5 min each), the blot was incubated with secondary antibody (IRDye 680RD goat anti-mouse antibody (Licor, #926-68070, Lot D10512-15), diluted 1:15000 in PBST). After another five wash steps (PBST, 5 min each), the blot was imaged with an Odyssey CLx fluorescence scanner (Licor). Data analysis was performed as described in the in-gel fluorescence section.

### AlphaFold model

For Fig. 1, an AlphaFold2 model of a complex between *M. mazei* Connectase and *M. mazei* recognition sequence (RELASKDPGAFDADPLV-VEI) was generated. This was done by gathering and aligning all available Connectase sequences from *Methanosarcinales* and *Methanomicrobiales*. The sequences were trimmed to the length of the

*M. mazei* variant. A GSGSGSG linker was added to the C-termini, as well as a region of the MtrA sequence from each organism. This region included the KDPGA motif and 15 residues N- and C-terminal of that motif. The structure of the *M. mazei* sequence within this dataset was then predicted with AlphaFold2, using the other sequences as a custom alignment for the process. For this, the Colab AlphaFold implementation with Jackhmmer (available under https://colab.research.google.com/github/sokrypton/ColabFold/blob/main/beta/AlphaFold_wJackhmmer.ipynb#scrollTo=pc5-mbsX9PZC) was used with standard settings. The best-scoring model from this prediction was in good agreement with the known co-structure of the respective *M. jannaschii* complex (PDB: "6ZW0"). It is shown in Fig. 1, without the linker and without 10 residues N-terminal of said MtrA region. (Note: The more conventional approach, i.e., the prediction of the complex from two separate protein chains, resulted in unconvincing models.)

### Statistics and reproducibility

No statistical method was used to predetermine the sample size. Three independent experiments ($n = 3$) were performed for the preparation of Figs. 4 and 8. Two independent experiments ($n = 2$) were performed for the preparation of Figs. 6 and S9 and S11. The experiments shown in Figs. 3, 5, S1, S2, S3, S4, S5, S10, S12 were performed only once. Although no exact replicates were made in these cases, the results are in agreement with a multitude of similar experiments used to develop the in-gel fluorescence method, including nearly identical precursor experiments. No data were excluded from the analyses. The experiments were not randomized. The investigators were not blinded to allocation during experiments and outcome assessment.

### Reporting summary

Further information on research design is available in the Nature Portfolio Reporting Summary linked to this article.

### Data availability

Source data are provided with this paper. The primary data have also been deposited in the Mendeley public repository (https://data.mendeley.com/datasets/3n47z3g33b). The crystal structure data used in this study are available in the PDB database under accession code "6ZW0". Source data are provided with this paper.

### Material availability

We freely distribute the materials required to perform the assays in this paper. The kits include Cy5.5-conjugated peptides (100 μg; sufficient for thousands of gels), plasmids for bacterial expression of Connectase, and a reference protein. Purified Connectase and reference proteins are also available. We appreciate feedback to further improve the method.

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

## Acknowledgements

We thank Andrei Lupas for discussions and continuous support. We thank Birte Hernandez-Alvarez for handling the eukaryotic cell cultures and providing CnTagged OmpLA protein. We also thank Victoria Sanchez (Core facility, MPI for Biochemistry, Martinsried) and Dai Long Vu (MPI for Biology, Tübingen) for mass spectrometric analyses. This work was supported by institutional funds from the Max Planck Society and by the German Research Foundation (DFG project number 512378754 to A.C.D.F.).

## Author contributions

A.C.D.F. designed, performed, and analyzed all experiments and wrote the paper.

## Funding

## Competing interests

Max Planck Innovation has filed a provisional patent on Connectase and its use for enzymatic ligation (EP-Patent Application 20807419; Inventors: Dr. Adrian Fuchs, Dr. Marcus Hartmann, and Dr. Moritz Ammelburg). The application covers all aspects of this work. The author declares no other competing interests.
