## [Peer Review File · Nature Communications]

REVIEWER COMMENTS

Reviewer #1 (Remarks to the Author):

This manuscript describes an elegant approach of using connectase ligation to link a fluorophore to tagged proteins, such that they can be detected on SDS-PAGE sensitively and quantified. I was impressed with the results obtained and the careful evaluation of linearity for different proteins. I think that this will have broad interest for the field. My main concern is that it is written much like an advert. The text conveys extreme enthusiasm for the method, with some of the claims not fully supported (see below) and a lack of balance in the description of alternative methods. There are too many such areas of lack of balance through the text for me to explain every one but here are a few:

(i) While the key precedent, ref 14 is cited, it is not made clear how that paper also displays many of the advantages of avoiding membrane transfer and direct in-gel detection.

(ii) "While commercial POI-specific antibodies are available for well-studied targets, no suitable antibodies are on the market for the majority of proteins."

It is rather more than well-studied proteins commercially available. See how many thousand antibodies are available from the leading suppliers.

(iii) If I understand correctly, the CnTag needs to be at the exact N-terminus. This is alluded to but should be made clearer. Various proteins need their native N-terminus for full activity.

I wonder if this will pose a problem for proteins with an N-terminal signal sequence in eukaryotic cells, where the Pro after the signal peptide might interfere with signal peptide cleavage?

The second important concern is that the Methods often lack detail.

e.g. buffer A- which acetate? What is the final pH?

Cells were lysed here without protease inhibitors, which is conventional to block degradation of many important cellular proteins. Please provide evidence if that affects the Connectase process?

The methods indicate that the samples are not heated at elevated temperatures after SDS addition (95C for at least 3 min is conventional). Is that essential for the procedure?

I would worry that some stable proteins may not unfold fully without a heating step.

The methods do not specify minimum or maximum time for gel fixation.

Spell out Licor settings, because that is fundamental to the whole paper.

A full gel (not just strips) should be shown in the main figures to illustrate how specific is the connectase approach on mammalian cell lysate.

Minor points:

Fig. 1: the chemistry of the process could be better explained

Fig. 3: molecular weight markers should be shown, at least on the Coomassie gel

“Labeling rates were only slightly decreased at 4 M urea (Figure S3), so that inclusion body proteins in urea may be detected with the same procedure as soluble proteins in physiological buffer.” I didn’t find the data on this use with inclusion bodies, in which case it should be made clear that this is a prediction.

Fig. S1C: puzzling spacing on x-axis

Fig. S6: only half the lanes are labelled. It could be clearer what the other lanes are.

Reviewer #2 (Remarks to the Author):

Re: In-gel fluorescence: A fast, sensitive and quantitative 1 alternative to Western blots

The manuscript was well written and provided significant supporting data to support the conclusions that the in-gel detection method presented in the manuscript can be used for similar analyses as Western blots with tag-specific antibodies, but has several added advantages. I was impressed by how quickly the CnT-tagged proteins could be visualized.

Some concerns:

The title is inadequate as it suggests that the manuscript developed a new method to detect any protein, including non-recombinant proteins. The title needs to be re-worded to be appropriate for the manuscript.

The data presented all compare the new method to western blotting but comparisons to already available in gel (nonwestern blotting) fluorescence detection, such as His12-tags were not carried out.

In figure 3 a light background band is seen in the line where 4 bands were detected (just below the middle of the gel).

No molecular weights are shown for the proteins detected in any of the figures.

Figure 4. the signal-to-substrate relationship is complex and should have been discussed in more detail. From 0-0.5 it may be linear then linear again from 1 to 8 fmol.

In figure 5. For IGF a 10-fold decrease in the amount of protein loaded did not seem to significantly decrease the levels of the recombinant proteins detected by in gel fluorescence (by eye).

In figure 6B. doublets are seen for some of the bands. What is the cause of the doublets and were both bands used in the quantification?

We would like to thank both reviewers for their comments and constructive criticism. All points that were raised are addressed in the revised manuscript (highlighted) and explained in detail below.

REVIEWER COMMENTS

Reviewer #1 (Remarks to the Author):

This manuscript describes an elegant approach of using connectase ligation to link a fluorophore to tagged proteins, such that they can be detected on SDS-PAGE sensitively and quantified. I was impressed with the results obtained and the careful evaluation of linearity for different proteins. I think that this will have broad interest for the field. My main concern is that it is written much like an advert. The text conveys extreme enthusiasm for the method, with some of the claims not fully supported (see below) and a lack of balance in the description of alternative methods. There are too many such areas of lack of balance through the text for me to explain every one but here are a few: (i) While the key precedent, ref 14 is cited, it is not made clear how that paper also displays many of the advantages of avoiding membrane transfer and direct in-gel detection.

We now make clear that this method allows direct in-gel detection (p. 1, ln 29-30). We also added a new paragraph, in which we discuss and compare published in-gel detection methods (p. 9, ln 30-41).

(ii) "While commercial POI-specific antibodies are available for well-studied targets, no suitable antibodies are on the market for the majority of proteins." It is rather more than well-studied proteins commercially available. See how many thousand antibodies are available from the leading suppliers.

We rephrased the sentence (p. 1, ln 21).

(iii) If I understand correctly, the CnTag needs to be at the exact N-terminus. This is alluded to but should be made clearer.

We now clarify that only N-terminal CnTags are studied and that C-terminal labeling requires a different procedure (p.2, ln 23-24).

C-terminal labeling is currently studied in our lab (see also p. 10, ln 10-11). The method is similar, but different substrates (e.g. PGAFDADPLVVEI-Cy5.5; POI-KDPGAFDADPLVVEI) and different concentrations (enzyme, substrate, fluorophore) need to be used.

Various proteins need their native N-terminus for full activity. I wonder if this will pose a problem for proteins with an N-terminal signal sequence in eukaryotic cells, where the Pro after the signal peptide might interfere with signal peptide cleavage?

We only studied on example so far, HER2. Here, we added the full Connectase recognition sequence, (linker)-kdpgafdadplvvei after the N-terminal signal peptide (residues 1-22). The detection works well, but the procedure is slightly different to what we describe in the paper. We do not know yet whether fusing the CnTag directly after residues 1-22 would also work with signal peptide cleavage. In any case, such applications need to be studied further before we are comfortable with publishing labeling

recommendations for such targets. In the meantime, we think it is clear that the CnTag just like any other protein tag may or may not interfere with protein functions.

In addition to the above (points i-iii), we made several changes to the text to discuss the method and alternatives more objectively and to add support to the arguments made. These changes include

- In several places, more careful expressions were chosen, e.g. p. 5, ln 17; p. 8, ln 22; p. 9 ln 2-4, ln 9, ln 17 and ln 27-28.
- We emphasized the importance of appropriate controls and validations (p. 9, ln 10; Assay protocol (supplement) and p. 6, ln 16-17).
- The suggestion to use two reference standards to increase the concentration range of the assay has been removed as this is not shown in the paper (p. 7, ln 23-26).

The second important concern is that the Methods often lack detail. e.g. buffer A- which acetate? What is the final pH?

The information was added (p. 11, ln 29-30). We also added more detail in other places.

Cells were lysed here without protease inhibitors, which is conventional to block degradation of many important cellular proteins. Please provide evidence if that affects the Connectase process?

We actually used lysis buffer with a common protease inhibitor mix. We added the composition of the lysis buffer to the methods (p. 11, ln 10-11). Connectase is not affected by common protease or proteasome inhibitors (Fuchs et al., PNAS 2021).

The methods indicate that the samples are not heated at elevated temperatures after SDS addition (95C for at least 3 min is conventional). Is that essential for the procedure? I would worry that some stable proteins may not unfold fully without a heating step.

We added information on heat incubation of the samples (p. 11, ln 39-40; p. 13, ln 40). This step is optional. It does not affect the signal as Connectase forms a regular (heat stable) amide bond between POI and fluorescent reagent and as the employed fluorophore Cy5.5 is heat stable. It is, however, only needed for proteins that do not unfold in loading buffer without heating (p. 11, ln 39-40).

The methods do not specify minimum or maximum time for gel fixation.

It is now stated that there is no minimum / maximum fixation time, but for prolonged storage fixation solution is recommended (p. 12, ln 2-3). Gels can be imaged immediately after the run (p.11, ln 42).

Spell out Licor settings, because that is fundamental to the whole paper.

Done (p. 10, ln 43-44).

A full gel (not just strips) should be shown in the main figures to illustrate how specific is the connectase approach on mammalian cell lysate.

We prepared a new figure (Figure 3B) showing that the method results in a high signal-to-noise ratio in several cell extracts (including mammalian cell extracts).

Minor points:

Fig. 1: the chemistry of the process could be better explained

We added additional information in the figure legend. However, many aspects of the reaction mechanism and the chemistry behind it are unknown so far, so we could not go further into detail.

Fig. 3: molecular weight markers should be shown, at least on the Coomassie gel

Molecular weight markers were added.

“Labeling rates were only slightly decreased at 4 M urea (Figure S3), so that inclusion body proteins in urea may be detected with the same procedure as soluble proteins in physiological buffer.” I didn’t find the data on this use with inclusion bodies, in which case it should be made clear that this is a prediction.

We added a Figure S5F, which shows that inclusion body proteins can be detected and quantified with the same procedure as soluble proteins.

Fig. S1C: puzzling spacing on x-axis

The points on the X-axis are the same as the ones on the gel. We now labeled all datapoints and adapted the description in the figure legend for clarity.

Fig. S6: only half the lanes are labelled. It could be clearer what the other lanes are.

We added labels to all lanes (now Figure S7).

Reviewer #2 (Remarks to the Author):

Re: In-gel fluorescence: A fast, sensitive and quantitative alternative to Western blots

The manuscript was well written and provided significant supporting data to support the conclusions that the in-gel detection method presented in the manuscript can be used for similar analyses as Western blots with tag-specific antibodies, but has several added advantages. I was impressed by how quickly the CnT-tagged proteins could be visualized.

Some concerns:

The title is inadequate as it suggests that the manuscript developed a new method to detect any protein, including non-recombinant proteins. The title needs to be re-worded to be appropriate for the manuscript.

We changed the title.

The data presented all compare the new method to western blotting but comparisons to already available in gel (nonwestern blotting) fluorescence detection, such as His12-tags were not carried out.

It was not trivial to do direct comparisons as the fluorophore-coupled super-chelator reagents for the detection of His12-tags¹ (hexaNTA-Alexa647) and His6-tags² (trisNTA-Alexa647) are not publicly available. One of the authors kindly offered to send us one of the reagents, but was not sure how well it would perform as it was already quite old at the time.

We did not want to use these methods under suboptimal conditions and therefore decided to replicate the relevant experiments in these studies, i.e. Brüchert et. al Figure 4a¹ and Raducanu et. al Figure 3². For this, we used the same CnTagged proteins (MBP and Sumo) and the same protein and extract quantities (Figure S4, lanes 1-8 in both gels). In addition, we also tested far lower protein quantities (lanes 9-15) and thereby show that Connectase-mediated in-gel fluorescence is more sensitive compared to these approaches (2000x more sensitive in case of Figure S4A; >200x more sensitive in case of Figure S4B (the detection limit is not reached here and may be ~5x lower)).

We did not replicate quantitative experiments, as such experiments were not carried out by Brüchert et. al¹, while Raducanu et. al² show a strongly convex signal-to-substrate relationship in the supplement (Figure S3), which is suboptimal for quantifications.

- 1 Bruchert, S., Joest, E. F., Gatterdam, K. & Tampe, R. Ultrafast in-gel detection by fluorescent super-chelator probes with HisQuick-PAGE. *Commun Biol* **3**, 138, doi:10.1038/s42003-020-0852-1 (2020).
- 2 Raducanu, V. S., Isaiglou, I., Raducanu, D. V., Merzaban, J. S. & Hamdan, S. M. Simplified detection of polyhistidine-tagged proteins in gels and membranes using a UV-excitable dye and a multiple chelator head pair. *J Biol Chem* **295**, 12214-12223, doi:10.1074/jbc.RA120.014132 (2020).

In figure 3 a light background band is seen in the line where 4 bands were detected (just below the middle of the gel).

The band corresponds to the small amount of residual N-Cnt reagent in the reaction. It is more intense in the control reaction without POI (lane 1). We clarified this in the text (p. 3 In 9) and in the figure legend.

No molecular weights are shown for the proteins detected in any of the figures.

We added the molecular weights and in all main text and supplement figures (except for strips showing a single protein).

Figure 4. the signal-to-substrate relationship is complex and should have been discussed in more detail. From 0-0.5 it may be linear then linear again from 1 to 8 fmol.

A closer look at the data suggests that the signal-to-substrate relationship is not linear in any region. Note that the X-axis shows a logarithmic scale. The curve on this scale is best described as sigmoidal with a half-maximum at ~3 fmol and a maximum at >25 fmol (p. 3, In 33-34 and p. 4 In 1-2). We now provide the primary data for this Figure along with manuscript for those that want to go into detail here. The main conclusion for the manuscript, however, is that all fluorophore is used in the reaction at POI quantities >25 fmol as this is the basis for the competition assay.

In figure 5. For IGF a 10-fold decrease in the amount of protein loaded did not seem to significantly decrease the levels of the recombinant proteins detected by in gel fluorescence (by eye).

The reason for this is that more protein of interest is available than fluorophore (see Figure 4 and the comment above). We describe this in the main text (p. 5, ln 12-15) and added a clarification in the figure legend.

In figure 6B. doublets are seen for some of the bands. What is the cause of the doublets and were both bands used in the quantification?

We found doublets only in Figure 6C, second gel. We think of them as a technical problem (perhaps varying SDS-gel quality) as they do not occur in other experiments with the same proteins (e.g. Figure 6C, first gel or Figure 3). We had the samples for the doublet-gel frozen and reapplied them on a new gel, which is now shown in the figure instead. No doublets are seen on this gel.

In addition, some gels show more or less pronounced shadows or smears (e.g. the reference band in Figure 6B on the second (but not on the first) gel). This may be a similar technical problem and also occurs on some Coomassie-stained gels. The assay is also accurate in these cases. We always used the whole signal (band+shadow) for quantification. This is now described on p.12, ln 1-2.

REVIEWERS' COMMENTS

Reviewer #1 (Remarks to the Author):

My comments have been addressed and the manuscript is substantially improved by these edits.

Reviewer #2 (Remarks to the Author):

The authors did a great job of addressing my major concerns.

Some minor concerns still need to be addressed.

Fig. 3. The units of molecular weight are missing. Please add kDa. This should be done for all figures with western blots.

Figs 4, 5, and 8. The molecular weight marker closest to the band or the molecular weight calculated size of the bands should be added, similar to figure 6B.

REVIEWERS' COMMENTS

Reviewer #1 (Remarks to the Author):

My comments have been addressed and the manuscript is substantially improved by these edits.

Reviewer #2 (Remarks to the Author):

The authors did a great job of addressing my major concerns.

Some minor concerns still need to be addressed.

Fig. 3. The units of molecular weight are missing. Please add kDa. This should be done for all figures with western blots.

Units (kDa) have been added to all figures showing gels and blots.

Figs 4, 5, and 8. The molecular weight marker closest to the band or the molecular weight calculated size of the bands should be added, similar to figure 6B.

Molecular weights have been added to Fig. 4, 5 and 8.